# Nutrition and Gut–Brain Pathways Impacting the Onset of Parkinson’s Disease

**DOI:** 10.3390/nu14142781

**Published:** 2022-07-06

**Authors:** Damiano Terenzi, Anne-Katrin Muth, Soyoung Q. Park

**Affiliations:** 1Department of Decision Neuroscience and Nutrition, German Institute of Human Nutrition (DIfE), Potsdam Rehbrücke, 14558 Nuthetal, Germany; anne.katrin.muth@gmail.com; 2Charité-Universitätsmedizin Berlin, Corporate Member of Freie Universität Berlin, Humboldt-Universität zu Berlin, and Berlin Institute of Health, Neuroscience Research Center, 10117 Berlin, Germany; 3Deutsches Zentrum für Diabetes, 85764 Munchen-Neuherberg, Germany

**Keywords:** Parkinson’s disease, nutrition, inflammation, biomarker, prevention

## Abstract

An emerging body of literature suggests that long-term gut inflammation may be a silent driver of Parkinson’s disease (PD) pathogenesis. Importantly, specific nutritive patterns might improve gut health for PD risk reduction. Here, we review the current literature on the nutritive patterns and inflammatory markers as a predictor for early detection of PD. This knowledge might be used to foster the detection of early nutritive patterns and preclinical biomarkers to potentially alter PD development and progression.

## 1. Introduction

Parkinson’s disease (PD) is the second most common neurodegenerative disorder after Alzheimer’s disease and is now the fastest-growing neurological disease worldwide in terms of prevalence and disability [1,2]. According to the Global Burden of Disease (GBD) study, incident cases of PD in 2017 were ~1.02 million, compared with ~2.5 million in 1990 [2,3,4]. PD is more prevalent in men than women [5,6].

Clinically, PD is defined as a progressive movement disorder, including bradykinesia, rigidity, and rest tremor [7,8]. Furthermore, PD is often characterized by different nonmotor symptoms affecting sensory perception, cognition, mood, motivation, autonomic functions, and sleep, among others [9,10,11,12,13]. 

Pathologically, the hallmark of PD is the aggregation of misfolded α-synuclein (aSyn) protein (a neuronal protein modulating neurotransmitter release), otherwise known as Lewy bodies. Lewy bodies are a primary cause of dopaminergic loss and related motor impairments in PD [14]. In particular, the development of α-Syn pathology in dopaminergic neurons within the substantia nigra pars compacta (SNc) [14], a brain region involved in modulating motor movement, is associated with the motor deficits (e.g., tremor and bradykinesia) observed in PD [15]. A sustained inflammatory response is a key pathological feature of PD. Microglia are a class of neuroglia (neuronal support cell) situated in the central nervous system (CNS). These cells are the immune cells of the CNS and consequently play a key role in neuroinflammation [16,17]. Human and animal postmortem studies, as well as positron emission tomography (PET) studies, have shown that there is a strong microglia activation in several regions of the PD brain, including the SNc and the striatum [16,17,18,19]. A prolonged microglia activation can have detrimental effects on the brain such as the increased expression of proinflammatory cytokines (small cell proteins involved in cell signaling such as interleukin-6 (IL-6) and interferon gamma (IFN-γ)) [20]. It has been suggested that these increased levels of cytokines contribute to the degeneration of the nigrostriatal DA neurons in PD [21]. Hence, excessive or misfolded α-Syn-induced neurotoxicity in PD may be partially mediated by altered microglia activity [20]. Astrocytes are glia cells contributing to the maintenance of brain homeostasis and neuronal metabolism. Like microglia, astrocytes respond to inflammatory stimulations and can themselves induce inflammation [22]. It has been proposed that PD may be partially due to astrocyte dysfunction [17,23]. In particular, α-Syn aggregation in PD can trigger microglia inflammatory responses. These proinflammatory mediators released by microglia can in turn activate astrocytes, which further magnify the inflammatory responses [20].

Regarding the etiology of PD, it is likely that, for most cases, the interplay between genetic and environmental factors contributes to the causation of the disease [24,25]. Among the environmental factors, there has been growing research interest on the relationship between the gut microbiota and PD [26,27,28,29,30,31]. The gut microbiota is the generic term referring to more than 100 trillion microbes (mostly bacteria but also viruses, fungi, and protozoa) that are present in the human gastrointestinal tract [32]. The accumulating evidence showing a bidirectional link between bacteria in the gut and neurons in the CNS (known as the “microbiota–gut–brain axis”) has resulted in an ongoing redefinition of health and disease, including neurodegenerative diseases [33,34]. In particular, recent studies have suggested that alterations in the composition of the gut microbiota (known as gut dysbiosis) might be a potential trigger for neuroinflammation, which in turn may lead to the development of PD [35,36,37,38]. In line with this hypothesis, α-Syn aggregation may start in the gut and then be spread from the gastrointestinal tract to the midbrain via the vagus nerve, resulting in the selective death of SNc [38,39]. Hence, alterations of the gut microbiota in PD may be used as an early biomarker of the disease [40,41,42]. Indeed, PD is often preceded by gastrointestinal symptoms, and gastrointestinal disorders accompany the disease [43]. Thus, testing blood and fecal biomarkers of gut inflammation and gut barrier permeability in PD might improve the accuracy of current clinical diagnostic criteria. 

Investigating nutritive patterns and their association with PD is a further interesting and emerging approach [44,45]. Increasing evidence suggests that specific nutritive patterns may influence (positively or negatively) the microbiota–gut–brain axis and, in turn, the risk of developing PD [44,46]. These nutritive patterns include dietary macronutrients (carbohydrates, proteins, and fats) [35,46,47], the intake levels of omega-3 (ω3) fatty acids [48], fruits, and vegetables [45,49], and the adherence to specific diets (e.g., Mediterranean diet or Western diet) [44,50].

In this review, we first discuss the relationship between gut dysbiosis and PD pathogenesis [51], focusing on several possible biomarkers of inflammation. Next, we review emerging studies suggesting a link between long-term nutritive factors and diet in adulthood with subsequent risk of developing PD.

## 2. Long-Term Gut Inflammation: A Silent Driver of Parkinson’s Disease Pathogenesis

### 2.1. Current Evidence on the Gut–Brain Hypothesis of Parkinson’s Disease

The CNS and the enteric nervous system (ENS) are connected through a bidirectional network, which is known as the gut–brain axis [52] (see Figure 1). This axis is crucial in maintaining homeostasis of both CNS and ENS, and it comprises multiple pathways of communication including endocrine (through the hypothalamic–pituitary–adrenal (HPA) axis) [53], immune (cytokines) [54], and neural (through the vagus nerve) [39] pathways. Regarding the neural pathway, the vagus nerve is one of the most direct pathways linking the gut and the brain and vice versa. This nerve, being part of the parasympathetic system, is characterized by both afferent (sensory, 80% of the fibers) and efferent (motor, 20% of the fibers) fibers [39,55]. Importantly, the gut microbiota can reach the CNS and, in turn, alter behavior through the vagus nerve. Indeed, microbiota metabolites can be detected through the afferent fibers of the vagus nerve. This gut information is in turn transferred to the CNS in order to produce a response [52,55]. Accumulating evidence suggests that this microbiome–gut–brain axis might play a key role in the underlying pathological mechanisms of PD [35,36,41].

At the beginning of the 2000s, human postmortem studies led by Braak and colleagues suggested for the first time that such α-Syn pathology is not confined to the CNS and is also detectable in the ENS [56,57,58], which is a division of the peripheral nervous system that controls the gastrointestinal system [59]. The evidence of α-Syn pathology also in this system lent support to the theory that PD pathology could be initiated in the ENS [36] and that it could be spread from the gastrointestinal tract to the midbrain via the vagus nerve, resulting in the selective death of SNc dopamine neurons [57,60] (see Figure 1). 

Growing experimental (in vitro and in vivo studies) and clinical evidence supports Braak’s hypothesis [61,62]. For example, as firstly observed by Braak and colleagues (2003), most PD patients (about 70% of patients) [63] develop gastrointestinal disorders such as constipation, dysphagia, and gastroesophageal reflux [43,56,64]. Strikingly, these gastrointestinal dysfunctions (particularly constipation and delayed gastric emptying) can be detected up to 20 years prior to PD diagnosis [37]. Furthermore, in line with Braak’s hypothesis on the role of the vagus nerve in spreading α-Syn aggregations from the gut to the brain [45], epidemiological studies have found that vagotomy can decrease PD risk [65,66]. Similar results have been found in animal models. Gastrointestinal disorders have been reported in animal models of PD at both early and advanced stages of the disease [67,68,69,70]. Furthermore, α-Syn aggregations were observed in animal models during both early and advanced PD [61,71,72]. Strikingly, truncal vagotomy and α-Syn deficiency prevented the spread of α-Syn aggregations from the gut to the brain and associated neurodegeneration and behavioral deficits in mice models of PD [60]. 

Despite this experimental and clinical evidence supporting Braak’s hypothesis, it may not accurately describe the development of PD in all patients [61,73]. Thus, future longitudinal studies are needed to investigate the disease progression, particularly in preclinical and prodromal stages of PD.

### 2.2. Inflammatory Biomarkers of Parkinson’s Disease

In an interesting study by Scheperjans et al. (2015), the authors compared the fecal microbiomes of 72 PD patients and 72 controls. Results suggested an altered microbiome in PD participants by showing a reduction in their number of Prevotellaceae bacteria compared to controls. Strikingly, this alteration was associated with the severity of the motor symptoms [29,35]. Further findings linking gut dysbiosis and PD pathogenesis showed that PD patients may exhibit not only a decreased abundance of *Prevotella* but also an increased abundance of Lactobacilliceae [35,74]. These alterations may be associated with reduced levels of ghrelin, an important gut hormone involved in the survival and efficacy of dopaminergic neurons [74]. Another study showed altered fecal microbiota in PD patients compared to healthy matched controls. In particular, the study found that an abundance of *Bacteroides* (Gram-negative bacteria present in the gut microbiome) is associated with the severity of motor symptoms in PD [30]. Lipopolysaccharide (LPS) is a biomarker of Gram-negative bacterial infection that can cause inflammatory responses [41,75]. Interestingly, studies have found high serum LPS levels in PD patients [41,76], which can reflect altered intestinal permeability already in the early stages of the disease [76]. Furthermore, several studies on animal models of PD have shown that the administration of LPS (stereotaxic, systemic, or intranasal) can reproduce the specific motor features of PD [41,76]. Similarly, other studies on PD have used LPS-binding protein (LBP), a protein that binds to bacterial LPS, finding lower plasma LBP levels in PD patients compared to healthy controls [42,77]. Overall, these findings support the role of LPS-elicited neurotoxicity in PD [41,77].

Other possible inflammatory markers of PD are calprotectin and zonulin. The first is a marker of inflammation, while the second is a junction protein, which, if modulated by proinflammatory signals (e.g., LPS), can lead to an increase in intestinal membrane permeability [78]. Interestingly, recent findings showed that both fecal and serum levels of calprotectin and zonulin are elevated in patients with PD [78,79,80]. In line with this evidence, animal models of dopamine degeneration and human studies have suggested that peripheral inflammation could contribute to the etiology and evolution of PD [81,82]. A substantial number of studies have started to investigate peripheral inflammation due to the easy access to blood samples. For example, it has been shown that PD patients may show altered levels of blood proinflammatory cytokines, which are signaling molecules released by immune cells such as helper T cells (Th) and macrophages. Specifically, enhanced levels of cytokines such as tumor necrosis factor alpha (TNF-α), interleukin 1 beta (IL-1β), interleukin 6 (IL-6), interleukin 10 (IL-10), and interleukin 8 (IL-8) have been found in PD patients when compared with healthy controls [82,83,84]. Similar to cytokines, chemokines (a family of small cytokines or signaling proteins) are also altered in PD. In particular, elevated levels of chemokine C–X3–C motif ligand 1 (CX3CL1) have been reported in PD patients when compared with controls [82]. Furthermore, two studies have reported significant elevated levels of C–X–C motif chemokine ligand 12 (CXCL12) in PD patients [82,85].

Overall, the evidence here reported suggests (see Figure 1) that gut dysbiosis and inflammatory processes/barrier dysfunction may facilitate the mechanism underlying dopaminergic neurodegeneration in PD. As mentioned in the previous section, symptoms such as constipation can precede the motor symptoms by even more than a decade; therefore, the investigation of the biomarkers related to gut dysbiosis might be particularly relevant in preclinical models of PD. However, prospective evidence is scarce [31,78,86], and it is still not clear whether gut dysbiosis is a cause or an effect of the disease. Thus, future research in preclinical and prodromal cohorts and longitudinal observations are still warranted. 

## 3. Nutritive Patterns as a Predictor of Early Detection of Parkinson’s Disease

Nutritional intake has been shown to define the healthy function of the CNS as a major lifestyle factor [47]. On the flip side, emerging studies suggest that neurodegenerative disorders such as PD may be partially due to the influence of unhealthy nutrition among other factors [44,45]. This raises the possibility of using dietary manipulations as a valuable strategy to preserve brain function and prevent neurodegeneration [45,50]. Different mechanisms may be related to the effect of nutrition on the development/progression of PD. For example, recent epidemiological findings have shown that some nutritive patterns can impact the vulnerability to oxidative stress and inflammation, which in turn may increase the risk of developing PD [35,46,87]. In contrast, other nutritive patterns may have neuroprotective effects that may decrease the risk of PD [50,88,89,90,91].

### 3.1. Mitochondria and Reactive Oxygen Species

Mitochondria are cell organelles that generate most of the cell’s energy through respiration and oxidative phosphorylation. The resultant energy is stored in adenosine triphosphate (ATP) molecules [92]. Thus, the mitochondria are involved in energy metabolism, stress response, and cell death [93]. Over time, mitochondrial DNA mutations and net productions of reactive oxygen species (ROS) accumulate. Specifically, ROS are a large family of oxidant molecules derived from the consumption and utilization of oxygen. Damaged mitochondria may lead to reduced ATP production and increased ROS accumulation, thereby contributing to aging [94]. Moreover, mitochondrial dysfunction plays a role in the etiology of PD [95] and might even be an early feature of the disease [96,97]. Mitochondrial dysfunction not only results in increased ROS levels and lower energy production [96] but also induces apoptosis, leading to degeneration of dopaminergic neurons [98]. In addition, other mitochondrial abnormalities are associated with PD, including mitochondrial electron transport chain impairment and changes in mitochondrial morphology and dynamics [99,100]. ROS play an important role in cellular and signaling pathways. However, excessive amounts that are not balanced by antioxidants contribute to oxidative stress [101], thereby leading to cellular degeneration [102] and cognitive decline [103]. In a recent paper, van Rensburg and colleagues (2021) proposed a toxic feedback loop that links uncurbed ROS and iron in the substantia nigra, due to aging, environmental exposure, and/or genetic predisposition [104]. Iron is needed for processes such as oxygen transportation, oxidative phosphorylation, myelin production, and neurotransmitter synthesis in the brain [105]. Accumulation of iron induces oxidative stress by generating ROS and can lead to apoptosis [106] and ferroptosis [107,108]; hence, it is implicated in PD [109]. Importantly [92,98,102], dietary intake can buffer or exacerbate the consequences of high levels of ROS [47], as further discussed in sections below.

### 3.2. Macronutrient Intake

Macronutrients—carbohydrates, dietary fatty acids, and proteins—impact cognitive functioning [110] and metabolic health via multiple pathways such as glucose metabolism and ROS levels associated with inflammation [47]. Such pathways are, in turn, linked with PD pathology [111,112,113] (see Figure 1).

Intake of specific macronutrients, as well as the diet’s macronutrient composition (i.e., the relative ratio of proteins, carbohydrates, and fatty acids), may affect several of these pathways [114,115,116], thereby linking dietary intake with PD risk. For instance, excessive protein intake has been shown to increase ROS in mice [117] while protein restrictions in rats reduced ROS damage in the liver [118]. Similarly, carbohydrate metabolism can impact ROS levels via glucose levels [119]. The effect of fatty acids on ROS depends on their type. For instance, high intakes of saturated fatty acids (SFAs) are proinflammatory [120], whereas polyunsaturated fatty acids (PUFAs) reduce ROS and have anti-inflammatory properties [121,122]. Accordingly, high omega-3 PUFA intake has been associated with lower risk PD risk [48,88,90,123], as well as with lower cognitive impairment, dementia, and depression [48,124], which are often comorbid with PD [45].

There is increasing scientific evidence on the possible efficacy of PUFA supplementation in slowing the cognitive and physical decline in PD [45,48]. However, randomized, double-blind, placebo-controlled clinical trials are scarce [124,125,126] and involve patients with an established diagnosis of PD, meaning that the pathology has already occurred, and neurons are compromised. Thus, this may limit the efficacy of possible dietary treatment in advanced stages of PD. 

### 3.3. Micronutrient Intake

Vitamin D serum levels are affected by both sun exposure and dietary intake (primarily from animal products and fortified foods). Inadequate vitamin D levels play a role in both chronic and neurodegenerative diseases, including PD [127], with sustained insufficiency playing a crucial role in PD pathogenesis [128]. Furthermore, vitamin D has been proposed to be a key driver of aging processes, including mitochondrial dysfunction, oxidative stress, and inflammation [129] (see Figure 1). In addition to these key processes that are affected in PD, lack of vitamin D also leads to central loss of dopaminergic neurons [127], as well as delayed gastric emptying in PD [127,130]. Two randomized controlled trials investigated the safety and efficacy of vitamin D3 supplementation in patients with PD [131,132]. The first study found that a treatment schedule of 1200 IU for 12 months was both safe and preventative of further deterioration in motor and nonmotor symptom domains using a total of 137 patients [131]. Another study investigated a supplementation schedule of 10,000 IU for 8 weeks, finding that younger participants (aged 52–66) benefitted in terms of motor symptoms, but older participants (aged 67–86) did not [132]. 

Other vitamins that have been studied in PD include B vitamins and antioxidant vitamins such as vitamins C, E, and A. Regarding vitamin B, studies have found that vitamin B6 is a critical cofactor for a wide range of biochemical reactions, including the synthesis of dopamine [133]. Therefore, it has been suggested that vitamin B6 may have a role in the development of PD. Accordingly, in a study by De Lau et al. (2006), vitamin B6 intake was associated with a reduced risk of developing PD. However, this result was observed only among smokers [134]. Indeed, some studies have found that smoking is associated with a reduced risk of developing PD as tobacco may regulate striatal activity through the dopaminergic system [135,136]. Thus, evidence linking vitamin B6 with the risk of developing PD should be interpreted with caution. Moreover, no associations have been found between the intake of other B vitamins such as B9 and B12 and the risk of developing PD [137]. Furthermore, studies examining the association between levels of vitamins C, E, and A and the risk of PD produced conflicting results. One study suggested that higher vitamin E intake may be associated with a reduced risk of developing PD [138]. However, no associations between the intake of vitamins C and E and incidence of PD were observed in another study [139]. Similarly, another study by Paganini and colleagues (2015) did not find associations between the intake of vitamins A and C and PD risk [140].

To sum up, according to the abovementioned studies, vitamin D may be protective against the development of PD. However, evidence regarding other vitamins intake and PD is mixed, and future studies are warranted. 

### 3.4. Dietary Patterns

Recent epidemiological studies have suggested that diet can increase or decrease the risk of developing PD via the microbiota–gut–brain axis [44,45,50] (see Figure 1). For example, it has been shown that the Western diet is one of the greatest risk factors for PD. This diet is usually characterized by a high intake of energy-dense foods, with a high content of proteins, saturated fat, refined grains, sugar, alcohol, and salt, as well as a reduced consumption of omega-3 (ω3) fatty acids, fruits, and vegetables [49,50,141]. Several foods that are part of the Western diet (e.g., beef, ice cream, fried foods, and cheese) have been associated with PD progression [49,50]. Furthermore, high total energy intake (another characteristic of the Western diet) was positively associated with the risk of developing PD in a meta-analysis including nine clinical studies [142]. It has been proposed that Western diet gut dysbiosis and altered intestinal barrier function can induce neuroinflammation [50,143,144,145]. In a study on a mouse model of PD, a high-energy diet such as the Western diet led to decreased parasympathetic functioning and α-Syn accumulation in the brainstem [146]. As mentioned earlier in this review, the α-Syn accumulation is, in turn, associated with the death of dopaminergic neurons in the SNc in PD. Moreover, a typical Western high-fat diet could increase insulin resistance, which in turn could impair nigrostriatal dopamine function in a rat model of PD [147]. This finding is in line with several clinical studies reporting that motor symptoms of PD are worse in individuals with comorbid type 2 diabetes [148,149,150]. Overall, the findings described above suggest that several mechanisms may contribute to the effects of Western diet on PD risk including neuroinflammation and insulin resistance. Importantly, gut dysbiosis and intestinal barrier alterations induced by the Western diet can increase blood levels of LPS (see Section 2.2). Altered levels of LPS can, in turn, activate Toll-like receptors (TLRs), a family of receptors that constitute a sort of immune system against bacteria [151]. Overstimulation of this system may provoke proinflammatory reactions, as well as enteric neuroglial activation, eventually eliciting α-Syn pathology [50,151].

Unlike the Western diet, the Mediterranean diet usually consists of plant-based foods, vegetables, legumes, fruits, nuts, seeds, fish, monounsaturated fats from olive oil, and whole grains [152,153]. Interestingly, the Mediterranean diet has been associated with a reduced risk of PD [88,91,154,155]. It has been suggested that this diet may improve intestinal barrier health and normalize insulin levels via the increased release of short-chain fatty acids (SCFA) (due to the high intake of fiber-rich foods—typical of the Mediterranean diet) [50]. All these factors may reduce neuroinflammation and, thus, the risk of developing PD [50,156]. Accordingly, it has been shown that PD patients have reduced production of SCFAs when compared to matched controls [156,157]. However, future randomized clinical trials are needed to confirm the role of the Mediterranean diet in reducing the risk of PD. 

The ketogenic diet (KD) is a high-fat low-carbohydrate diet that has been used as a nonpharmacologic approach to improve a range of health markers in metabolic disorders [158,159,160,161], as well as a treatment for intractable epilepsy [158,162,163,164]. In recent years, studies have suggested that the KD can be therapeutically useful for adjunctive therapy for PD [165]. In particular, this diet may have neuroprotective effects. The KD consists of a very high fat content with very little carbohydrate intake and normal to low protein intake. The exact ratios depend on the specific type of KD. Metabolically, KD increases ketone bodies and reduces oxidative stress brought on by excessive ROS [46]. Evidence from animal models of PD showed promise, as ketone bodies acted neuroprotectively [166] and improved motor skills [167]. Evidence in humans is limited to two studies. The first was a feasibility study in five PD patients who consumed a KD for 4 weeks [87]. All patients showed a decrease in the Unified Parkinson Disease Rating Scale (UPDRS) total scores; however, without a control group, these findings cannot be interpreted [87]. More recently, a randomized controlled trial followed 47 patients with PD that were assigned either to a 1750 kcal KD or to a low-fat diet lasting 8 weeks each. Participants in the KD diet group showed significant improvements on the nonmotor daily living experiences part of the MDS-UPDRS, as well as a nonsignificant trend for improvement on the motor examination part compared to the low-fat diet group. The authors concluded that KD shows promise complementary to L-dopa treatment [168]. Although these promising results suggest that the KD may have beneficial effects in PD, the main obstacles are the adherence to the diet and the short- and long-term effects [46]. Importantly, results from larger randomized clinical trials have not yet been reported [158], and future studies are needed. 

## 4. Discussion and Conclusions

The current review provides an initial understanding of the role of the gut–brain axis in the development of PD. At the same time, this review offers a broader view of the systemic consideration of how nutrition may play a significant role in gut health and inflammation, with studies showing that specific nutritive patterns can increase or decrease the risk of developing PD. In particular, this review evaluates current evidence regarding the possible therapeutic utility of altering the gut microbiota through diet as a promising approach to prevent or modify PD progression. 

Firstly, we discussed the increasing evidence suggesting that nonmotor symptoms such as gastrointestinal dysfunctions may precede PD diagnosis by more than a decade, lending support to Braak’s hypothesis that PD pathology may start in the gut and then spread via the vagus nerve to the brain. In particular, we reviewed and discussed evidence from epidemiological studies and animal models supporting this hypothesis. We also consider how this hypothesis may not accurately describe the development of PD in all patients. Hence, we suggest that future longitudinal studies are needed to investigate PD progression, particularly in preclinical and prodromal stages of the disease.

Secondly, we showed how the possibility that PD pathology may start in the gut is also supported by recent discoveries of close links between PD and inflammatory biomarkers. Accordingly, we discussed the current literature on these biomarkers by focusing on fecal and blood markers of gut dysbiosis in PD. More in detail, we reported that gut dysbiosis revealed by Prevotellaceae, Lactobacilliceae, *Bacteroides*, LPS, calprotectin, and zonulin levels emerged as some of the most consistent altered biomarkers in PD. Furthermore, peripheral inflammation has been revealed in PD by measuring the levels of blood inflammatory cytokines (e.g., TNF-α, IL-1 β, IL-6, IL-10, and IL-8) and chemokines (e.g., CX3CL1 and CXCL12). Overall, the evidence here reported suggests that gut dysbiosis and related inflammatory processes/barrier dysfunction may facilitate the mechanism underlying dopaminergic neurodegeneration in PD. We suggest that further prospective longitudinal studies could help to possibly identify alterations in these biomarkers already in the early stages of PD. Importantly, the many crucial biomarkers discussed in this review have the potential to improve the diagnosis of PD. However, the different biomarker profiles may not only vary significantly across people but could also change during the different stages of the disease. Future studies should test the accuracy of combined biomarkers for a more precise diagnosis and treatment of PD [169]. 

Thirdly, we summarized the evidence showing the impact of nutrition on the gut–brain axis and its possible role in the development of PD. Among different nutritive patterns, we focused on specific macro- and micronutrients, as well as on different dietary patterns, which have been recently shown to be possible modulating factors of neurodegeneration in PD. 

Specifically, regarding the macronutrients, we discussed the role of carbohydrates, dietary fatty acids, and proteins and their impact on cognitive functioning and metabolic health in PD, possibly via multiple pathways such as glucose metabolism and ROS levels. In particular, we discussed the increasing scientific evidence on the possible efficacy of PUFA supplementation in slowing the cognitive and physical decline in PD. Furthermore, we summarized and discussed recent data suggesting that vitamin intake (particularly vitamin D) may be associated with the risk of developing PD. 

Lastly, we revised several prospective longitudinal studies showing associations between dietary patterns and the risk of PD. Collectively, we reported that the Western diet has been associated with an increased risk of developing PD, possibly via diet-induced neuroinflammation and insulin resistance. Conversely, the Mediterranean diet (characterized by a high intake of dietary fibers) may improve intestinal barrier health and normalize insulin levels via the increased release of SCFA. All these factors may reduce neuroinflammation and, thus, the risk of developing PD. We also discussed recent human and animal evidence showing that the ketogenic diet has promise for treating PD, as ketones may have neuroprotective effects by reducing oxidative stress. 

To conclude, the gut–brain axis discussed in this review is a hot topic in the study of PD. Over the past 5 years, studies testing the gut–brain hypothesis in the etiology and progress of PD have markedly expanded, and several systematic reviews and meta-analysis on this topic have been published [27,35,36,37,38,41]. However, as a future direction, much more work is needed to be able to develop new therapeutic approaches that may act on the gut microbiota composition such as probiotic or dietary therapies, as well as hopefully even preventative approaches. Regarding probiotics, these are living microbes that can have anti-inflammatory or antioxidant properties. Some recent studies have suggested that probiotics can represent a good dietary intervention for neurodegenerative diseases such as PD and Alzheimer’s disease (AD) [170,171,172]. Specifically, probiotics may promote an increase in anti-inflammatory factors and a decrease in proinflammatory cytokines [172,173], thereby contributing to reducing intestinal inflammation in PD [172]. However, findings from clinical trials in neurodegenerative diseases are inconsistent [170,173,174,175,176]. Additionally, evidence on the probiotic mechanisms of actions came mostly from animal studies [172,177], and future studies are needed to investigate their effects in humans [172]. In addition to the use of probiotics as a potential therapy in PD, as mentioned earlier, in this review, we highlighted areas where future research on the effect of nutritive patterns on PD is highly relevant. In particular, the role of the Mediterranean diet and the use of PUFA supplementation in PD should be further investigated as they may positively influence the onset of motor and nonmotor symptoms of the disease. 

## Figures and Tables

**Figure 1 nutrients-14-02781-f001:**
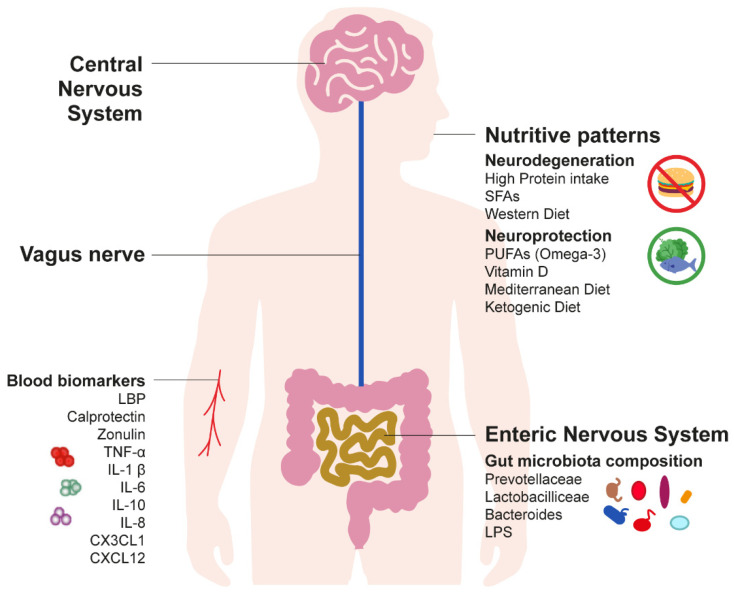
Schematic representation of the microbiota–gut–brain axis in the pathogenesis of Parkinson’s disease (PD). Alterations in the gut–microbiota composition (gut dysbiosis) may affect the aggregation of alpha-synuclein (α-Syn). The spread of α-Syn pathology from the enteric nervous system (ENS) to the central nervous system (CNS) is mediated by the vagus nerve. Gut dysbiosis may be revealed by Prevotellaceae, Lactobacilliceae, *Bacteroides*, and lipopolysaccharide (LPS). Possible blood biomarkers of gut dysbiosis and related inflammation in PD are LPS-binding protein (LBP), calprotectin, zonulin, tumor necrosis factor alpha (TNF-α), interleukin 1 beta (IL-1β), interleukin 6 (IL-6), interleukin 10 (IL-10), interleukin 8 (IL-8), C–X3–C motif ligand 1 (CX3CL1), and C–X–C motif chemokine ligand 12 (CXCL12). The nutritive patterns that may be associated with neurodegeneration in PD are the excessive protein and short-chain fatty acid (SFAs) intake, and the adherence to the Western diet. Conversely, a high intake of polyunsaturated fatty acids (PUFAs) and vitamin D, as well as diets such as the Mediterranean diet and the ketogenic diet, may have neuroprotective effects in PD.

## Data Availability

Not applicable.

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
