# Peer review of "Nutrition and Gut–Brain Pathways Impacting the Onset of Parkinson’s Disease"

_nutrients, 2022, doi:10.3390/nu14142781_

Round 1
Reviewer 1 Report
In the manuscript titled "Nutritive pattern and inflammatory marker as a predictor for early detection of Parkinson’s disease" Damiano Terenzi and colleagues, they have reported that One of the pitfalls of Parkinson’s disease (PD) treatment is the late detection, at the stage when the motor or cognitive symptoms have already appeared. An emerging body of literature suggests that long-term gut inflammation may be a silent driver of PD pathogenesis. Importantly, specific nutritive patterns might improve gut health for PD risk reduction. Here, we review the current literature on the nutritive pattern and inflammatory marker as a predictor for early detection of PD. This knowledge might be used to foster the detection of early nutritive patterns and pre-clinical biomarkers to potentially alter PD development and progression. I have a few comments regarding the present manuscript.
-The introduction section needs more sentences regarding nutritive patterns and inflammatory markers.
-Definition of gut-brain axis and how microbiota could influence this are required.
-In my opinion is not necesaary sumarizes in every section the principal information, instead of summarizes at end.
-Maybe a figure thank summarizes the information is required
-I read the entire manuscript, but the focus of the present review maybe is missing. A reestructuration maybe is needed. How and why the nutrition have impact in the Parkinson's disease, and later focus in sub specific headings.
-Definitions are missing in the entire document.
Reviewer 2 Report
-The review must be implemented with Figures and Table summarrising the current literature, for example on the different nutrients/micronutrients intake and their effects.
- Even if the focus is nutrition , the pathological features of PD, and its inflammatory nature should be underlied in deep.
Reviewer 3 Report
The present review aims to discuss the relation between nutritive bioactives, inflammatory markers and PD onset and progression. Overall, the ideas discussed in the review are good, but the text is not well structured and there are shortcomings from the point of view of discussion and consistency between the various topics covered in the sections. Furthermore, there are more points that should be clarified and some speculative phrases that should be better discussed and deepened.
Specific comments
Title
The title should be revised as the nutritive patterns can indeed influence the onset of PD but it is speculative to say that they can predict PD as well. In the same way, the Authors emphasize inflammation markers but in the text there’s very little about them.
Abstract
Line 11: This first sentence is completely out of scope with respect to what is stated later in the abstract
Introduction
Line 50 - this sentence has nothing to do with the Introduction section, it should be discussed later in the other sections
Lines 47-51 these sentences have been used also in the abstract and in the Discussion and Conclusion section, therefore they should be modified or removed
2. Long-term gut inflammation: a silent driver of Parkinson’s disease pathogenesis
Line 54 - as with the other sections, it is redundant for me every time to introduce the topic of the section as done in this review
Line 59 – please state what ENS stands for
2.2. Inflammatory biomarkers of Parkinson’s disease
Line 85 – same as for Line 54
Line 98 – “another relevant biomarker of inflammatory…”. Actually, LPS is not a biomarker of inflammation but, instead, of Gram-negative bacteria infection. Anyway, it would be the first component relating to inflammation mentioned in the text, because previously it was only talked about modulation of the cmicrobiota composition
Line 108 – “other possible inflammatory markers”… in reality, except for the LPS, no others have been mentioned but the composition of the fecal microbiota. This section should be implemented with more information on other biomarkers
Line 109 - saying that zonulin is a marker of intestinal permeability is not quite correct. If anything, it is true that zonulin is a junction protein which, if modulated by proinflammatory signals (e.g. LPS), can lead to an increase in membranes permeability. This fact should be better discussed
3.1. Mitochondria and Reactive Oxygen Species
Line 131 - this topic is very important and should be treated with more emphasis, while as it is structured it makes little sense and is not useful for the reader
4. Discussion and Conclusions
This section has been written taking exactly the information shown in the previous sections (starting with Line 262), summarizing what had already been described. In my opinion this section should well highlight and explain the most important findings of this collection of evidences, but above all it should also discuss future scenarios regarding the topic and the shortcomings with respect to the research topic itself. For this reason, I think this section should be substantially revised, because a mere repetition of what has already been written is not useful for the reader and lowers the quality of the review.
Line 304 - this sentence seems unsuitable for the conclusion of the review and should be better argued anyway. Moreover, it has also been used in the Introduction section, so it is redundant
Round 2
Reviewer 1 Report
Thanks to the authors for including all my comments in the present manuscript. Reading again the entire document, now reads well and no further comments are required from my side. All the best.
Reviewer 2 Report
THE MANUSCRIPT HAS BEEN IMPROVED AND IT IS SIUTABLE FOR PUBLICATION
Reviewer 3 Report
As for me, the authors have answered exhaustively to all requests and suggestions. The paper as such can be published in Nutrients